# Efficient and bright warm-white electroluminescence from lead-free metal halides

Hong Chen[1,7], Lin Zhu[1,7], Chen Xue [2,7], Pinlei Liu[1], Xuerong Du[1], Kaichuan Wen[1], Hao Zhang[1], Lei Xu[1], Chensheng Xiang[3], Chen Lin[4], Minchao Qin [5], Jing Zhang[6], Tao Jiang[1], Chang Yi[1], Lu Cheng[1], Chenglong Zhang[1], Pinghui Yang[1], Meiling Niu[1], Wenjie Xu[1], Jingya Lai[1], Yu Cao [1,2], Jin Chang[1], He Tian [3], Yizheng Jin [4], Xinhui Lu[5], Lang Jiang [6], Nana Wang [1✉], Wei Huang [1,2✉] & Jianpu Wang [1✉]

Solution-processed metal-halide perovskites are emerging as one of the most promising materials for displays, lighting and energy generation. Currently, the best-performing perovskite optoelectronic devices are based on lead halides and the lead toxicity severely restricts their practical applications. Moreover, efficient white electroluminescence from broadband-emission metal halides remains a challenge. Here we demonstrate efficient and bright lead-free LEDs based on cesium copper halides enabled by introducing an organic additive (Tween, polyethylene glycol sorbitan monooleate) into the precursor solutions. We find the additive can reduce the trap states, enhancing the photoluminescence quantum efficiency of the metal halide films, and increase the surface potential, facilitating the hole injection and transport in the LEDs. Consequently, we achieve warm-white LEDs reaching an external quantum efficiency of 3.1% and a luminance of 1570 cd m$^{-2}$ at a low voltage of 5.4 V, showing great promise of lead-free metal halides for solution-processed white LED applications.

[1] Key Laboratory of Flexible Electronics (KLOFE) & Institute of Advanced Materials (IAM), Nanjing Tech University (NanjingTech), Nanjing, China. [2] Shaanxi Institute of Flexible Electronics (SIFE), Xi'an Institute of Biomedical Materials & Engineering (IBME), Northwestern Polytechnical University (NPU), Xi'an, China. [3] China Center of Electron Microscope, State Key Laboratory of Silicon Material, School of Material Science & Engineering, Zhejiang University, Hangzhou, China. [4] Center for Chemistry of High-Performance and Novel Materials, State Key Laboratory of Silicon Materials, and Department of Chemistry, Zhejiang University, Hangzhou, China. [5] Department of Physics, The Chinese University of Hong Kong, Shatin, Hong Kong, China. [6] Beijing National Laboratory for Molecular Sciences, Key Laboratory of Organic Solids, Institute of Chemistry, Chinese Academy of Sciences, Beijing, China. [7] These authors contributed equally: Hong Chen, Lin Zhu, Chen Xue. ✉email: iamnnwang@njtech.edu.cn; iamwhuang@nwpu.edu.cn; iamjpwang@njtech.edu.cn

Metal-halide perovskites have received considerable attention due to their unique properties, for example solution processibility and good optoelectronic properties[1–3]. After the rapid progress in the past few years, the efficiency of single-color lead-halide perovskite LEDs is approaching those of the best-performing organic LEDs[4–6]. However, efficient white perovskite LEDs remain a significant challenge. Mixing single-color perovskite material with organic compound[7,8] and stacking perovskite layers with various colors[9] have been used to achieve white lead-halide perovskite LEDs, but the peak EQE is only 0.22%[9]. Alternatively, various metal halides possessing broadband white-light emission, such as two-dimensional lead halide perovskite[10–12], lead-free halide double perovskite[13], and cesium copper halides[14], have been developed to be emitters in white LEDs. However, there is no report of electroluminescence (EL) efficiency from those materials yet and the maximum luminance of the device is only around 70 cd m$^{-2}$[10], which mainly due to their unfavorable electronic properties[13,14], or low photoluminescence quantum efficiencies (PLQEs)[10]. Among them, solution-processable cesium copper iodides seem promising for white LED applications, because they usually have high PLQEs with broad visible emission from self-trapped excitons or reorganized excited states, and relatively good air stability[15–20]. However, the electronic properties of the cesium copper iodides are unfavorable for LEDs since the charge transport is poor due to the large effective mass of carriers[14], and the charge injection is difficult due to the large bandgaps[15,21]. Here we find that the optoelectronic properties of the cesium copper iodides can be significantly enhanced by simply chemisorbing ether groups onto the metal-halide surfaces acting as electron donors. Based on this strategy, we first demonstrate efficient and bright warm-white EL based on lead-free halides.

## Results

**Efficient and bright LEDs based on lead-free metal halide.** The emitting layers were prepared by spin-coating a precursor solution of cesium iodide (CsI), copper(I) iodide (CuI), and polyethylene glycol sorbitan monooleate (Tween) with a molar ratio of 1:1:0.006 dissolved in dimethyl sulfoxide (DMSO) (14 wt.%) (see "Methods" section for details). The X-ray diffraction (XRD) data show that the as-prepared film is a mixture of zero-dimensional $Cs_3Cu_2I_5$ and one-dimensional $CsCu_2I_3$ (Fig. 1a). As shown in Fig. 1b, the peaks at 12.3°, 15.1°, 23.9°, 24.7°, 25.5°, 26.2°, 27.0°, 28.1°, and 30.6°, correspond to the $Cs_3Cu_2I_5$ phase[16], and the peaks at 10.6°, 13.4°, 26.0°, and 29.2°, associate with the $CsCu_2I_3$ phase[17]. The PL spectrum shows a broad and white-light emission range from ~390 to ~740 nm (Fig. 1c), with a Commission Internationale de l'Eclairage (CIE) color coordinate of (0.42, 0.47). There are two distinct peaks at 437 and 570 nm in the PL spectrum, which can be assigned to the emission from excited-state structural reorganization and self-trapped exciton of the $Cs_3Cu_2I_5$ and $CsCu_2I_3$, respectively[15,17]. This assignment is consistent with the longer transient PL lifetime at 440 nm (1100 ns) and shorter transient PL lifetime at 578 nm (60 ns) (Supplementary Fig. 1a)[15,17]. The PL excitation (PLE) spectra measured at different emission wavelengths show that the PL emissions of $Cs_3Cu_2I_5$ and $CsCu_2I_3$ crystals stem from their absorption peaks at 287 and 314 nm (Fig. 1d and Supplementary Fig. 2a) respectively, consistent with the absorption spectra[15,16]. Notably, the optical properties of the film are very stable in the air. After keeping the deposited film in the air for over 1500 h, there is no degradation of the PL intensity (Supplementary Fig. 1b), consistent with previously reported good stability of $Cs_3Cu_2I_5$ and $CsCu_2I_3$ in the literature[15,17]. Importantly, with Tween in the precursor, both XRD, PL, and PLE peak positions of cesium copper iodides are identical to those without Tween (Fig. 1c and Supplementary Fig. 2), indicating that the Tween did not

change the crystal structure of the mixture. However, we can observe that both the XRD and PL peak intensities are enhanced with Tween. The results are consistent with the literature that the non-ionic surfactant Tween molecules in precursor solution can tune the crystallization and reduce the defects of $CsPbBr_3$ perovskites[22]. Scanning electron microscope (SEM) measurement also shows that the film with Tween forms a discrete film with larger grains than that of without Tween (Fig. 1e and Supplementary Fig. 2c). This leads to a higher PLQE of 30% than 18% in the control sample. These results suggest that the Tween can facilitate the growth of high-quality $Cs_3Cu_2I_5$ and $CsCu_2I_3$ crystals, and we will explain the reason below. In addition, the cross-sectional scanning transmission electron microscope (STEM) image shows Tween forms a ~9 nm organic insulating layer between the discrete cesium–copper–halide crystallites (Supplementary Fig. 3), which can prevent the leakage current in LED device[4]. And the discrete structure can carry over to top layers, which is beneficial for light outcoupling in LEDs[4,23].

The LED device structure is ITO/poly (3,4-ethylenedioxythiophene):poly (styrenesulfonate) (PEDOT:PSS, 30 nm)/metal halides (~60 nm)/1,3,5-Tri(*m*-pyridin-3-ylphenyl) benzene (TmPyPB, 45 nm)/lithium fluoride (LiF, 1 nm)/aluminum (Al, 100 nm) (Fig. 2a) (see "Methods" section for detailed fabrication process). The device shows a broad EL spectrum with a peak at 565 nm and a full width at half maximum (FWHM) of 121 nm (Fig. 2b), which has a CIE color coordinate of (0.44, 0.53) and a correlated color temperature of around 3650 K in the warm-white region[24]. The device maintains unchanged EL spectra at different bias voltages, exhibiting good color stability. The LED turns on at a low voltage of 2.7 V (corresponding to a luminance of 1 cd m$^{-2}$). After that, the current density and luminance increase quickly. These features are very different from the literature[13,15,21], suggesting the carrier injection and transport properties are significantly enhanced in our lead-free metal-halide LED device. The device shows a brightness up to 1570 cd m$^{-2}$ at a low voltage of 5.4 V and external quantum efficiency (EQE) of 3.1%, representing the first efficient and bright warm-white LEDs based on lead-free metal halide (Supplementary Table 1). Moreover, the device shows decent reproducibility, with an average EQE of 1.8%. We highlight that the impressive LED performance mainly benefits from the inclusion of Tween. Without Tween, the device performance is very poor, with a brightness of 35 cd m$^{-2}$ and a peak EQE of 0.04% (Supplementary Fig. 4), which is similar to the literature[15,21]. Furthermore, as changing the ratio of CsI:CuI to 1:2 to form a $CsCu_2I_3$-rich film (Supplementary Fig. 5a), the LED device exhibits a large leakage current and very low luminance due to the poor film morphology (Supplementary Fig. 5b, c, f). On the contrary, when increasing the ratio of CsI:CuI to 3:2 to form a $Cs_3Cu_2I_5$-dominant film, the unfavorable energy level alignment and low carrier mobility of $Cs_3Cu_2I_5$ would also reduce the EQE of device[15].

**The roles of additive containing ether group in cesium copper iodides.** The scanning Kelvin probe microscope (SKPM) measurements show that the metal-halide film with Tween has a higher and more uniform surface potential (contact potential difference) than the control film (Fig. 3). The average surface potential of the Tween sample is 248 mV, while that of the control film is only 10 mV. This shift in surface potential is in good accordance with the values obtained by ultraviolet photoelectron spectroscopy (UPS), which shows a change in work function from 4.62 to 4.17 eV after adding Tween (Supplementary Fig. 6a), indicating a shallower Fermi level. In addition, the valence band with respect to the Fermi level is shifted from 0.84 to 0.90 eV compared with the control sample, leading to the valence band

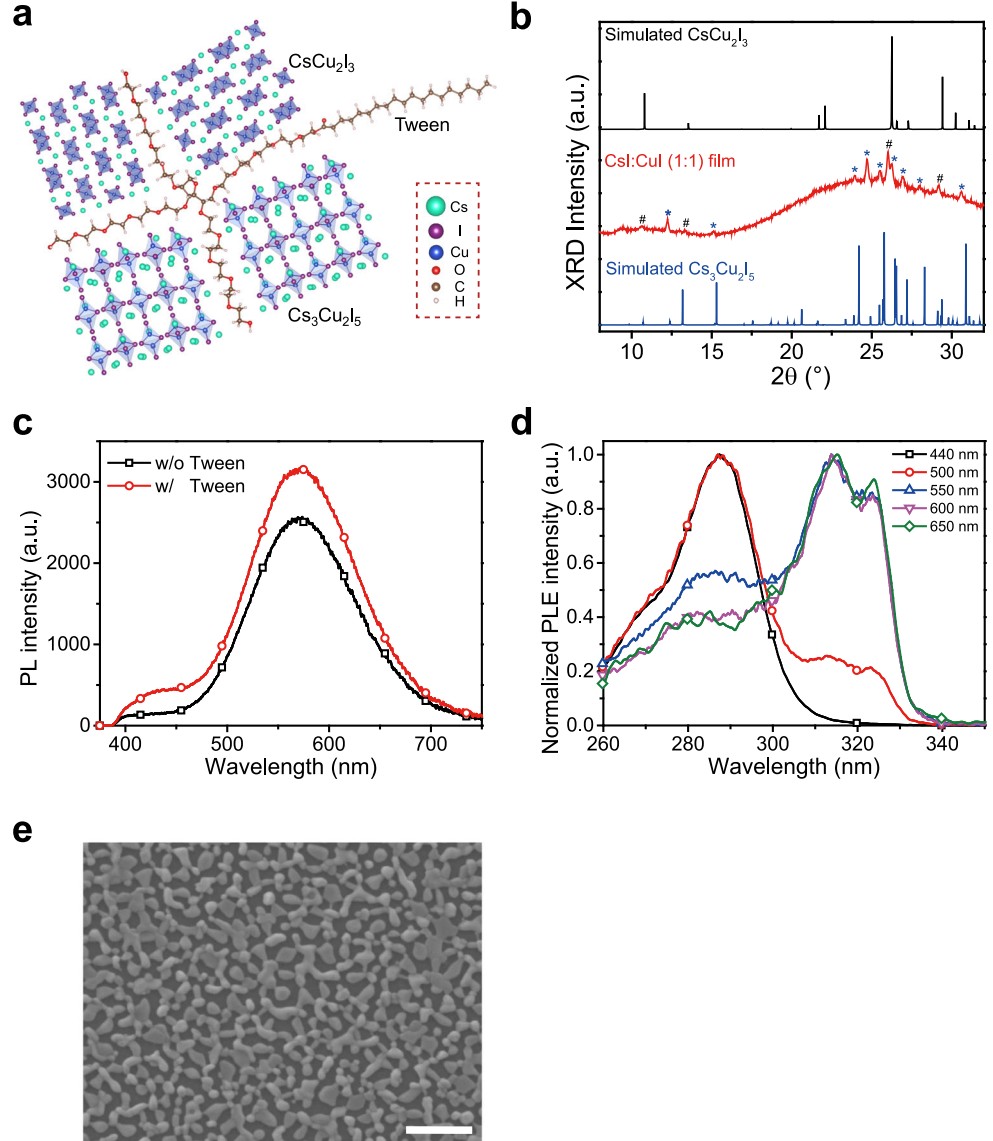

**Fig. 1 Characterizations of cesium–copper–iodide films with Tween. a** Schematic diagram of the emitter material, which is a mixture of $Cs_3Cu_2I_5$ and $CsCu_2I_3$ crystallites. The Tween molecule can chemisorb onto cesium copper iodides. **b** XRD pattern compared to the calculated XRD patterns of $Cs_3Cu_2I_5$ and $CsCu_2I_3$. * and # denote the diffraction peaks corresponding to $Cs_3Cu_2I_5$ and $CsCu_2I_3$. **c** PL spectra. The inclusion of Tween enhances the PL intensity. **d** Normalized PLE spectra at various emission wavelengths. **e** SEM image. Scale bar, 1 μm.

shifted by 0.39 eV toward the vacuum. A schematic flat-band energy level diagram for our devices is shown in Supplementary Fig. 6b, although there are likely complex band-bending and interactions that will occur close to the interfaces. However, the diagram is able to show that Tween can reduce the energetic barrier for hole injection owing to the shallow valence-band energy level of the emitting layer. After the inclusion of Tween, the hole current increases over ten times, as confirmed by the current density–voltage (J–V) measurement of the hole-only device (Supplementary Fig. 6c). We note that normally the hole mobility in cesium copper iodides is much lower than electron mobility[14], which has been a major issue to achieve efficient LEDs. In our case, the current of hole-only device is over two orders of magnitude higher than that of the electron-only device (Supplementary Fig. 6d). Apart from the enhanced hole current, this can also result from the difficulty in achieving electron injection due to the large conduction band offset between the emitting layer and the electron-transporting TmPyPB layer (Supplementary Fig. 6b). Therefore, we believe that the charge

recombination in the metal halides must happen near the interface of TmPyPB, and the enhanced hole injection and transport is critical to reduce the turn-on voltage and increase the current density of the LED devices (Fig. 2c).

Next, we further investigate why Tween can enhance the electronic properties of the cesium–copper–iodide films. To reveal the interaction between Tween molecules and cesium copper iodides, we performed Fourier transform infrared (FTIR) spectroscopy measurement. We can observe peaks at 1122, 2866, and 2924 cm$^{-1}$ in Tween film (Fig. 4a), which are attributed to the C–O–C and C–H stretching vibrations, respectively. After adding CsI to Tween, the relative intensity of C–H stretching vibration peaks changes, and the C–O–C absorption moves to lower wavenumber at 1107 cm$^{-1}$. Similar results are also observed in the Tween:CsI:CuI sample. The FTIR result suggests that there is a chemical interaction between Cs and the C–O–C bond of Tween. This interaction is further confirmed by X-ray photoelectron spectroscopy (XPS) measurements. Figure 4b shows that the Cs 3d and I 3d peaks of Tween:CsI film are

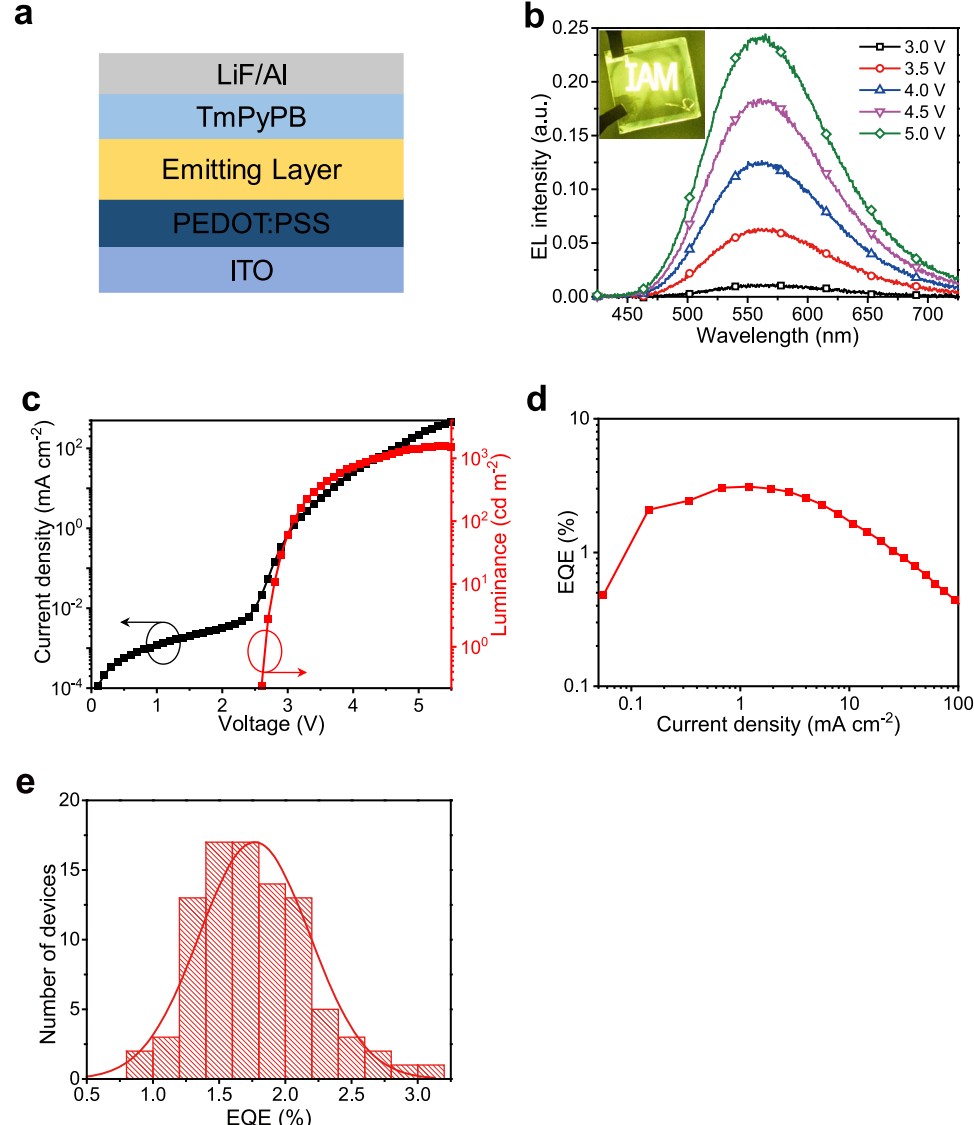

**Fig. 2 Characterizations of cesium–copper–iodide LEDs with Tween. a** Schematic diagram of the device structure. **b** EL spectra of the device under different voltages. Inset, the photograph of LED with the logo of IAM. The substrate area is 12 mm × 12 mm. **c** Dependence of current density and luminance on the driving voltage. Before turning on, the device shows low current density, indicating suppressed leakage current by the thin organic layer between crystallines[4]. After that, the luminance increases quickly and reaches a luminance of 1570 cd m$^{-2}$ at a low voltage of 5.4 V. **d** EQE versus current density. The LED reaches a peak EQE of 3.1%. **e** Histogram of peak EQEs. Statistics of 91 devices show an average peak EQE of 1.8% with a relative standard deviation of 23.5%.

shifted by 0.3 eV towards lower binding energy compared to that of the pristine CsI film, while the peak associated with O 1s in Tween moves from 532.8 to 533.0 eV in the Tween:CsI sample (Supplementary Fig. 7a). Similar shifts are also observed in the CsI:CuI sample with Tween. On the contrary, the Tween additive has no effect on the CuI spectra (Supplementary Fig. 7b). This opposite shift of Cs 3d and O 1s peaks is consistent with the formation of chemical bonds between Cs ions and oxygen atoms.

Density-functional theory calculation (see "Methods" section for details) shows that Tween can modify the crystallographic facets of CsCu$_2$I$_3$ and Cs$_3$Cu$_2$I$_5$ through the interaction between oxygen and Cs ions (Supplementary Fig. 8a, b)[25,26]. Notably, as the large spatial distribution of Cs$^+$ in low-dimensional structure allows the offset of Cs$^+$ within the lattices[27,28], the calculation shows beside the charge transfer between Tween and inorganic crystal through isolated Cs ions, the iodide to copper charge transfer and *d-s* transitions also occur within the Cu–I clusters (Supplementary Fig. 8c, d). This

indicates that Tween can enhance the charge transport not only on the surface but also in the bulk of crystals[29,30], which is consistent with the enhanced current density in electron- and hole-only devices with Tween (Supplementary Fig. 6c, d). Furthermore, the calculation shows that after the inclusion of additive, the work functions of CsCu$_2$I$_3$ and Cs$_3$Cu$_2$I$_5$ are reduced by 0.47 eV (from 4.01 to 3.54 eV) and 0.41 eV (from 5.12 to 4.71 eV), respectively, corresponding with the SKPM and UPS measurement results (Fig. 3 and Supplementary Fig. 6a). In addition, the additive can eliminate the sub-gap states caused by the Cs$^+$ and the dangling bonds on the exposed surface (Supplementary Fig. 8e, f), which indicates the oxygen atoms from the additive can saturate the dangling bonds and coordinate with the exposed Cs$^+$, leading to reduced trap states and enhanced PL efficiency.

Moreover, we find that the chemical interaction between Tween and Cs can also directly influence the crystallization process of the cesium copper iodides. We carried out synchrotron-based in situ

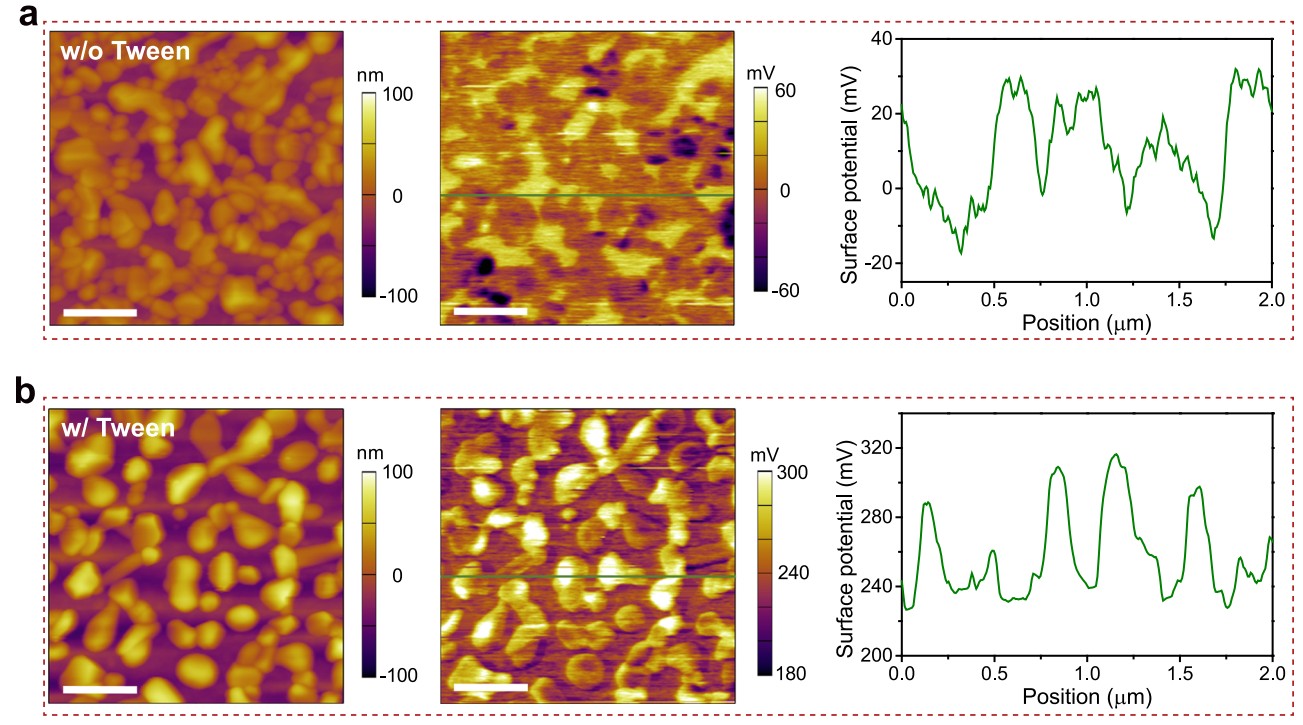

**Fig. 3 Height (left), surface potential (middle) images, and corresponding line scans of cesium–copper–iodide films.** Scale bar, 500 nm. **a** Without Tween. **b** With Tween. The inclusion of Tween increases the average surface potential of cesium–copper–iodide film. The surface potential of the control film ranges from −99 to 69 mV, and that of the film with Tween is from 183 to 326 mV.

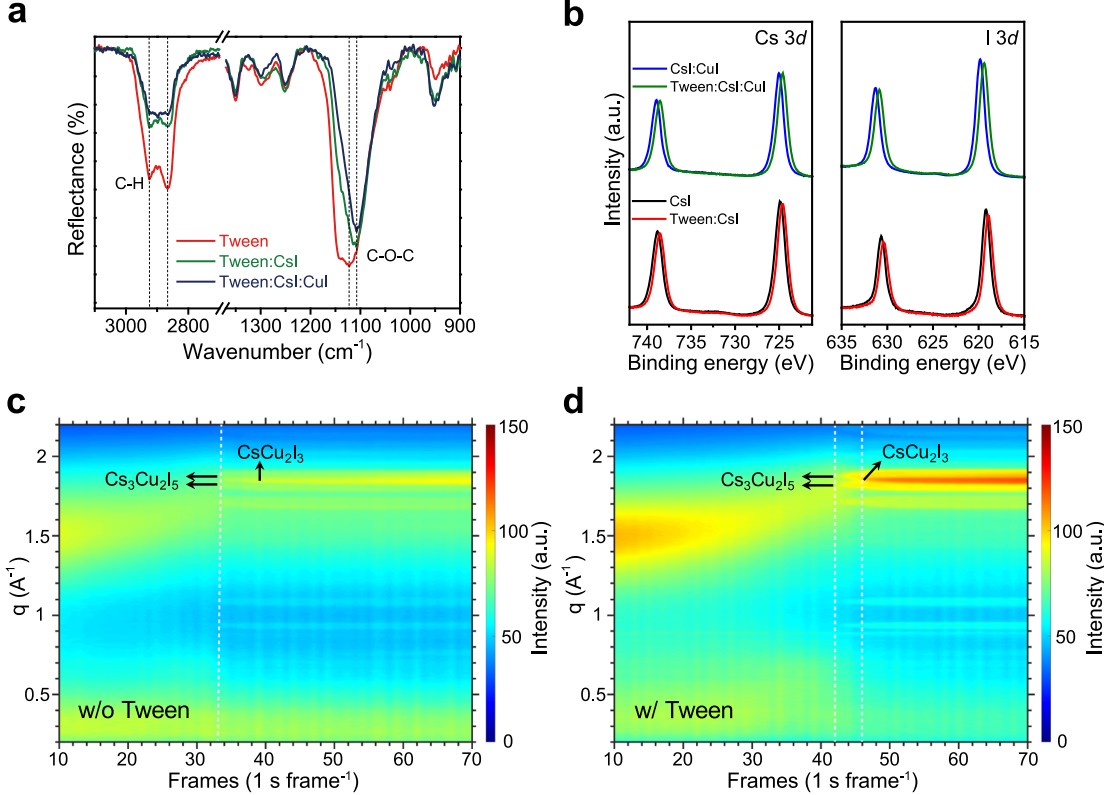

**Fig. 4 Characterizations of interaction between Tween and cesium copper iodides. a** FTIR spectra. The spectrum of Tween shows peaks at 2866 and 2924 cm$^{-1}$ ascribed to the C–H stretching vibrations, and a peak at 1122 cm$^{-1}$ ascribed to C–O–C stretching vibration. After adding Tween to CsI, the relative intensity of 2866 and 2924 cm$^{-1}$ peaks changes, and the 1122 cm$^{-1}$ peak shifts to 1107 cm$^{-1}$. **b** XPS spectra of CsI, Tween:CsI, CsI:CuI, and Tween: CsI:CuI films. After adding Tween to CsI, the Cs 3$d$ and I 3$d$ peaks of CsI shift to the lower binding energy. **c, d** Time-resolved GIWAXS profiles of samples without (**c**) and with (**d**) Tween. The sample without Tween simultaneously forms CsCu$_2$I$_3$ at |$q$| = 1.84 Å$^{-1}$ and Cs$_3$Cu$_2$I$_5$ at |$q$| = 0.93, 1.08, 1.71, 1.82, and 1.88 Å$^{-1}$ (around 33 s). After the inclusion of Tween, the CsCu$_2$I$_3$ appears at 46 s, which is 4 s later than the Cs$_3$Cu$_2$I$_5$.

grazing-incidence wide-angle X-ray scattering (GIWAXS) measurements during the spin-coating process. Figure 4c, d and Supplementary Fig. 9 show that there forms $Cs_3Cu_2I_5$ and $CsCu_2I_3$ almost at the same time (33 s) for the sample without Tween. In contrast, due to the electrostatic interaction between Tween and CsI, the $Cs_3Cu_2I_5$ forms at around 42 s in Tween-based halides, and then the $CsCu_2I_3$ appears 4 s later. This suggests that the chemical interaction between Tween and Cs can retard the nucleation of cesium copper iodides during the spin-coating process, which leads to films with enhanced crystallinity (Fig. 1b and Supplementary Fig. 9). We note that similar additive-enhanced crystallinity has been observed in lead halide perovskites[31].

The above experimental and simulation results consistently suggest the chemical interaction between the C–O–C bond in Tween and Cs is the key to the fabrication of efficient Cs–Cu–I LED devices through multiple roles, which include enhanced crystallinity, increased surface potential, enhanced charge transport and reduced trap states of the Cs–Cu–I films as well as working as insulators in the forming island structures to prevent leakage current. To test the generality, we fabricated the LEDs based on polyethylene oxide (PEO) additive, which is a polyether compound and shows similar chemical interaction with Cs, as confirmed by FTIR measurement (Supplementary Fig. 10a). Interestingly, the resulted PEO-based LED also exhibits improved peak EQE of 1.7% and maximum luminance of 890 cd m$^{-2}$, without further optimization of the device fabrication process (Supplementary Fig. 10b, c). We believe that the performance of our cesium–copper–iodide LEDs could be further enhanced by optimizing device structure to reduce the electron injection barrier, and introducing new additives to passivate defects and boost the PLQEs as demonstrated in lead-based perovskites[4,25].

## Discussion

It is interesting to observe that introducing foreign organic molecules into the precursor solution of cesium copper halides can result in profound changes in the optoelectronic properties of the formed metal-halide films. This change can be understood in terms of enhanced crystallinity, charge transfer, increased surface potential, and reduced defect states. Solution-processed efficient and bright warm-white LEDs based on the cesium copper iodides were demonstrated by using this simple approach. Considering the variety of both organic molecules and metal halides, we believe that the approach we developed can be extended to explore rich optoelectronic devices based on metal halides.

## Methods

**Device fabrication and characterization**. PEDOT:PSS (Clevios P VP 4083) was spin-coated onto ITO-coated glass substrates as a hole-transporting layer. The precursor solution of cesium copper halides was prepared by dissolving CsI, CuI, and polyethylene glycol sorbitan monooleate (Tween 80) with a molar ratio of 1:1:0.006 in DMSO (14 wt.%). The metal-halide films were prepared by spin-coating the precursor solution onto the PEDOT:PSS films and annealed at 100 °C for 5 min. Finally, the TmPyPB, LiF and Al electrodes were thermally evaporated. The substrate size is 12 by 12 mm$^2$, and every single device area is 3 mm$^2$. The measurement of LEDs was carried out using an integration sphere in a nitrogen-filled glovebox[32,33].

**Film characterizations**. All the films were prepared on PEDOT:PSS substrate as in the device. The SEM images were collected with a JEOL5 JSM-7800F SEM. The STEM images of ITO/PEDOT:PSS/metal halides/Au device were obtained on an FEI Titan G2 80–200 ChemiSTEM operated at 200 keV.

The absorbance spectra were measured by using a UV-vis spectrophotometer with an integrating sphere (PerkinElmer, Lambda 950). The PL and PLE spectra were obtained by using a Fluorescence spectrometer (Hitachi, F7100). A fluorescence spectrometer with an integrating sphere (Edinburgh Instruments, FLS980) was used to measure the PLQE[34]. The films were excited by a 285 nm excitation beam from a xenon lamp. Time-resolved PL measurements were carried out by using a spectrometer (Edinburgh Instruments, FLS980), with 280 and 310 nm pulsed lasers.

XRD patterns were performed using a Bruker D8 Advance. FTIR spectra were collected by using a Thermo Scientific Nicolet iS50 with a reflection accessory.

UPS spectra were recorded using a multifunctional photoelectron spectrometer (AXIS ULTRA DLD) with a He I ultraviolet radiation source (21.2 eV). XPS spectra were collected on a Thermo Scientific K-Alpha$^+$. The films were prepared on SiO$_2$/PEDOT:PSS substrates and all the XPS measurements were calibrated using the C 1s line. The SKPM measurements were carried out by using Asylum Research Cypher S. To eliminate test errors, all SKPM measurements were performed with the same probe.

The in situ GIWAXS measurements were conducted at the National Synchrotron Radiation Research Center (NSRRC), Hsinchu[35]. The wavelength of the X-ray was 1.240 Å (10 keV) and the sample to detector distance was calibrated with a lanthanum hexaboride (LaB$_6$) sample. The incident angle was 1° and a C9728DK area detector was used to collect the scattering signals. After the perovskite precursor was dropped on the substrate, concomitant GIWAXS measurements and sample spinning could be triggered simultaneously, and the spin-coating process was conducted in an air-tight chamber under N$_2$ flow.

**Single-crystal growth and characterization of $Cs_3Cu_2I_5$ and $CsCu_2I_3$**. The precursor solution of $Cs_3Cu_2I_5$ crystals was prepared by dissolving 0.9 mmol CsI and 0.5 mmol CuI in 300 μL DMSO. The crystals were grown by a slow vapor saturation of the antisolvent method[36]. The precursor solution was placed inside a jar filled with MeOH as an antisolvent. After keeping at room temperature for 3–4 days, $Cs_3Cu_2I_5$ crystals with a size of about 5 mm were obtained. The $CsCu_2I_3$ crystals were prepared using a similar method except for the 0.5 mmol CsI.

Single-crystal XRD data were obtained by using a Bruker APEX-II CCD diffractometer (Mo Ka radiation, $\lambda = 0.71073$ Å). Data reduction and absorption corrections were performed with the SAINT and SADABS software packages, respectively. Structures were solved by direct methods using the SHELXL-2014 software package. The atoms were anisotropically refined using a full-matrix least-squares method on $F^2$.

**First-principles calculations**. All calculations were performed by using the density-functional theory (DFT) and the projector-augmented wave (PAW)/plane-wave method as implemented in the Vienna Ab initio Simulation Package[37–39]. The Perdew, Burke, and Ernzerhof (PBE) functional is used[40]. Calculations were carried out with plane-wave basis using a plane-wave kinetic energy cutoff of 400 eV, and with a uniform $4 \times 4 \times 7$ $k$-point grid for $CuCu_2I_3$ (010) bulk model, $4 \times 1 \times 7$ $k$-point grid for $CuCu_2I_3$ (010) surface model and $4 \times 4 \times 1$ k-point grid for $Cs_3Cu_2I_5$ (001) surface model. Four formula units are employed for all the calculations. We started with experimental-determined crystal structures and carried out the structural relaxations until the changes of total energy reaching 0.05 eV/Å. The total energy is converged to within $1 \times 10^{-5}$ eV for each electronic self-consistent loop.

We used a four-layer slab with $1 \times 1$ surface supercell for $CsCu_2I_3$ (010) surface (10.48 × 6.06 Å) and $Cs_3Cu_2I_5$ (001) surface (10.10 × 11.57 Å). Two repeated functional part –OCH$_2$CH$_2$O– of Tween terminated with methyl and hydrogen was simplified as an additive to investigate the interaction between the additive and Cu-based compounds. A vacuum layer of 20 Å is used to separate images along the surface normal direction.

The adsorption energies were calculated by the following equation:

$$E_a = E_{adsorbed} - E_{decoupled} \qquad (1)$$

where $E_a$, $E_{adsorbed}$, and the $E_{decoupled}$ represents the adsorption energy, the energy of the adsorbed configuration, and the total energy of two distinct part without any interaction in the adsorbed configuration.

The calculations show that the surface energies for Cs-terminated (010) surface and Cu-terminated (010) surface are 0.453 and 5.564 eV Å$^{-2}$ respectively, suggesting the exposure of (010) surface terminated Cs atoms is more stable than Cu atoms and there should be a strong interaction between Cs and Tween. Therefore, the calculations only consider the preferred interaction between Tween and Cs site.

## Data availability

The data that support the finding of this study are available from the corresponding author upon reasonable request.

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

## Acknowledgements

This work is financially supported by the Major Research Plan of the National Natural Science Foundation of China (91733302), the National Natural Science Foundation of China (61875084, 61922041, 21601085, 51703094, 61961160733, 61905109), the National Science Fund for Distinguished Young Scholars (61725502), the Natural Science Foundation of Jiangsu Province, China (BK20180085, BK20170991, BK20171002), the Synergetic Innovation Center for Organic Electronics and Information Displays, the High Performance Computing Center of Nanjing Tech University, the beam time and technical support provided by 23A SWAXS beamline at NSRRC, Hsinchu, the financial support from Research Grant Council of Hong Kong (RGC) (14314216), N.W. is a Marie Skłodowska-Curie Fellow (841454).

## Author contributions

J.W. had the idea for and designed the experiments. J.W., W.H., and N.W supervised the work. H.C., X.D., L.C., and W.X. carried out the device fabrication and characterizations with the assistance of N.W. P.L. and T.J. contributed to the characterization of single-carrier devices. X.D., M.N., L.X., C.Y., and K.W. conducted the optical measurements. H.C., P.Y., J.C., and L.Z. carried out the XRD measurement. H.Z., C.Z., and Y.C. carried out the SEM measurements. C.L. measured SKPM under the supervision of Y.J. H.C., C.Y., and L.Z. carried out the FTIR characterizations. Chen S.X. carried out STEM measurements and H.T. supervised this characterization. Chen X. carried out the calculation of materials. M.Q. carried out the GIWAXS measurements under the supervision of X.L. J.Z. conducted the UPS measurements under the supervision of L.J. J.L. and Chen X. performed the XPS measurement. J.W., N.W., and H.C. analyzed the data. N.W. wrote the first draft of the manuscript. J.W. and W.H. provided major revisions. All authors discussed the results and commented on the manuscript.

## Competing interests
The authors declare no competing interests.
