## [Peer Review File · Nature Communications]

REVIEWER COMMENTS

Reviewer #1 (Remarks to the Author):

In this study, authors developed lead-free white LED using metal halide emitting materials, Cs₃Cu₂I₅ and CsCu₂I₃. Authors used the additive of Tween to reduce the trap states. Consequently, the white LED reached a EQE of 3.1% and the brightness of 1,570 cd/m². I believe that the reported EL performance is quite good among the lead-free LEDs to date. However, I feel that the present manuscript is not suitable for the publication in Nature communication. I think there are too many inconsistencies in scientific explanation. Besides, clear motivation and originality should be presented. Moreover, more scientific discussion is needed. Please see the details below.

1. In figure S1, I think the shown PL decay curves is not originated from the trap states related defects. The decay curve shapes rapidly change after a certain amount of time. I think this phenomenon is attributed to the chemical degradation owing to air-ambient. Noted that the CsCu₂I₃ is weak in air-ambient. It seems that the tween passivates the surface of CsCu₂I₃ and prevent the chemical reaction with O₂ or H₂O. But, this data does not prove that the surface trap states are passivated by Tween.

2. As seen in Fig. s6, authors performed DFT calculation to explain the role of Tween. However, only this calculation results cannot support the role of Tween. It should be noted that all inorganic materials possess surface states since owing to kinds of dangling bonding. In this case, some adsorption on surface generally stabilize the surface state. In other words, if authors do the calculation using the other organic molecules, the similar results will be obtained. If so, the role of Tween is still mysterious. Therefore, authors should further investigate on a variety of additives to confirm whether the Tween is specialized for Cs-Cu-I systems or not.

3. As authors know, there have been already many reports on the Tween additive for perovskite LEDs. Authors should clarify more the originality of this work.

4. In figure S5b, authors compared the energy levels between w/o Tween and w/ Tween. Ionization potential (I.P.) is intrinsic value of materials. As authors said, if there is difference in I.P., there should be some dipole moment. On the other hand, owing to the dipole moment the conduction band minimum (electron affinity) with Tween become shallow by 0.5 eV, implying that electron injection barrier height from ETL to Cs-Cu-I becomes larger. But, the data for electron only device indicates that electron injection barrier height is similar between w/o Tween and w/ Tween. This point is clearly scientific contradiction.

Reviewer #2 (Remarks to the Author):

This submission reports efficient electroluminescence from lead-free copper halide materials, which can be employed to fabricate LED devices. Following the emergence of lead halide perovskite solar cells, metal halides are currently attracting global interest for a variety of optical and electronic applications. In particular, a number of metal halides (both perovskite type and non-perovskite) have recently been shown to exhibit very high efficiency light emission properties. Given the strong interest of a broad community of researchers in luminescent metal halides, the topic of this submission is suitable Nature Communications.

This is an interesting work reporting a substantial improvement of LEDs based on copper iodides. The primary value of this work is the device improvement (i.e., engineering of better devices). I enjoyed reading the paper and I recommend its publication after a revision. In particular, the specific role of additive needs to be clarified better. This and other fundamental questions and suggestions are provided below:

1. "Among them, solution-processable cesium copper iodides seem promising for white LED applications, because they have high PLQEs with broad visible emission from self-trapped excitons or reorganized excited states, and good air stability." A few comments here: (i) The authors do not

cite a few related papers on cesium copper halides, e.g., [ACS Appl. Electron. Mater. 2019, 1, 3, 269–274] and [ACS Materials Lett. 2019, 1, 4, 459–465]. Air-stability of Cs₃Cu₂I₅ does not seem so good based on a recent publication [ACS Appl. Electron. Mater. 2019, 1, 3, 269–274], any comments? (ii) Do all copper halides demonstrate broad PL emission? Some do not seem so broad, FWHM < 100 nm for powder samples is not so broad? (iii) Some preliminary device work on CsCu₂I₃ has already been reported including in ref. [ACS Materials Lett. 2019, 1, 4, 459–465], in which the device architecture seems to imply the use of organic additives, too. Can the authors compare their results with the literature results and include in Table S1?

2. In Fig. 1b (also Fig. S2b), why the XRD pattern range is only between 20-30 deg? Also, what are the film thicknesses here, the diffraction peak intensities seem quite low?

3. In Fig. 1e, the film coverage seems poor. What are the darker spots between the grains? If these are Tween formations, is the primary role of Tween then prevention of formation of pinholes? The current explanation of the role of Tween seems to be focused on improved transport properties, which can be a side effect, too (i.e., if film coverage is poor, transport will also be poor).

4. The exact role of Tween remains confusing in this work. The authors seem to argue about a strong interaction between Tween and cesium copper iodides. What are the HOMO and LUMO energies of Tween and how do they compare to HOMO/LUMO (VB/CB) of Cs₃Cu₂I₅ and CsCu₂I₃? If the transport properties are improved, are the authors arguing that Tween helps with charge transport? If yes, please, clarify. Or is it simply a passivation of surface defects with Tween that helps the device performance?

5. The current best device seems to be mostly yellow emitting CsCu₂I₃ and the reported CIE coordinates seem further away from the white point. Can the authors optimize Cs₃Cu₂I₅/CsCu₂I₃ ratio to get closer to the white point? And more generally, how does changing the Cs₃Cu₂I₅/CsCu₂I₃ ratio impact the EQE of the resultant LEDs?

Reviewer #3 (Remarks to the Author):

The authors report the use of the Tween additive in Cs-Cu-I and successfully fabricate white LEDs with record performance in the lead-free halide family. Based on XRD, SEM, FTIR and GIWAXS, detailed analysis and calculations were carried out, which help explain the unique optoelectronic properties. This is an interesting and important work on the lead-free halides research. I have some comments which the authors need to address well:

(1) The XRD was only measured in a very short range between 20 and 30°. The authors need to measure from 5 to 40° and also include the simulated patterns in Figure 1b for comparison.

(2) From the SEM image, the thin film looks like isolated Cs₃Cu₂I₅ and CsCu₂I₃ were formed. In this case, do the LED devices have current leakage in the longitudinal direction? This requires careful consideration and discussions in the manuscript.

(3) The authors claim that the film is very stable in air (over 1500h). How stable are the LED devices?

(4) The white emission should be caused by the different photoluminescence behaviors of Cs₃Cu₂I₅ and CsCu₂I₃. Have the authors tried films with different ratios of CsI and CuI? If the PL changes with different starting materials, it will confirm the origin of the white light.

(5) I'm concerned about the statement of the role of Cs⁺ and Tween. Compared with Cs, Cu element has greater electronegativity. Tween should also play a role on Cu site.

Point-by-Point Response to Referees

Reviewer #1:

Comment #1: In this study, authors developed lead-free white LED using metal halide emitting materials, $\text{Cs}_3\text{Cu}_2\text{I}_5$ and CsCu_2I_3 . Authors used the additive of Tween to reduce the trap states. Consequently, the white LED reached a EQE of 3.1% and the brightness of $1,570 \text{ cd/m}^2$. I believe that the reported EL performance is quite good among the lead-free LEDs to date. However, I feel that the present manuscript is not suitable for the publication in Nature communication. I think there are too many inconsistencies in scientific explanation. Besides, clear motivation and originality should be presented. Moreover, more scientific discussion is needed. Please see the details below.

Response: We thank the reviewer for recognizing our device performance in lead-free LEDs.

Firstly, we confirm the data in Supplementary Fig. 1a and Supplementary Fig. 6b,d are consistent in scientific explanation. The detailed information can be found in response to the Comments #2 and 5.

Secondly, we have added more explanations of the originality of this work, which can be found in response to the Comment #4.

Thirdly, we have added more discussions of the role of Tween. The detailed information can be found in response to the Comment #3.

Comment #2: In figure S1, I think the shown PL decay curves is not originated from the trap states related defects. The decay curve shapes rapidly change after a certain amount of time. I think this phenomenon is attributed to the chemical degradation owing to air-ambient. Noted that the CsCu_2I_3 is weak in air-ambient. It seems that the tween passivates the surface of CsCu_2I_3 and prevent the chemical reaction with O_2 or H_2O . But, this data does not prove that the surface trap states are passivated by Tween.

Response: We thank the reviewer for this comment. The reason that the PL decay curves in the Figure S1 seem abnormal is due to the logarithmic scale of the x -axis. Actually, when changing the x -axis to linear scale as usual (see the below figure), the PL decay curves of both at 440 nm and 578 nm emission wavelengths are consistent with literatures (*Adv. Mater.* 2018, 30, 1804547; *Adv. Mater.* 2019, 31, 1905079). The big difference of the decay lifetime between the two emission

wavelengths can be attributed to the different emissions in $\text{Cs}_3\text{Cu}_2\text{I}_5$ (440 nm) and CsCu_2I_3 (578 nm), which are from the excited-state structural reorganization and self-trapped exciton respectively and have been widely investigated in literatures (*Adv. Mater.* 2018, 30, 1804547; *Adv. Mater.* 2019, 31, 1905079). We have revised the Supplementary Fig. 1a with linear scale of x -axis in the manuscript to make this point clear.

Time-resolved PL decay curves of CsI:CuI (1:1) films with or without Tween.

Comment #3: As seen in Fig. s6, authors performed DFT calculation to explain the role of Tween. However, only this calculation results cannot support the role of Tween. It should be noted that all inorganic materials possess surface states since owing to kinds of dangling bonding. In this case, some adsorption on surface generally stabilize the surface state. In other words, if authors do the calculation using the other organic molecules, the similar results will be obtained. If so, the role of Tween is still mysterious. Therefore, authors should further investigate on a variety of additives to confirm whether the Tween is specialized for Cs-Cu-I systems or not.

Response: We thank the reviewer for this comment. We have calculated the effect on the first layer in CsCu_2I_3 crystal by using organic molecules with other groups (i.e. CH_2 , NH) to substitute the O atom in Tween (see the below figure a). Firstly, it shows different binding energies between various additives and the CsCu_2I_3 (010) surface. We note that the CH_2 -substituted additive molecule has smaller interaction with the inorganic surface than Tween. This is consistent with our previous results that the O atom in Tween interacts with Cs ions (Supplementary Fig. 7a,b), which is also confirmed by the FTIR and XPS measurements (Fig. 4a,b). Secondly, the charge density difference of

additive/CsCu₂I₃ indicates that the charge redistribution mainly occurs at the CO/CsCu₂I₃ interface region, and there is almost no charge change on CsCu₂I₃ after the CH₂- or NH-substituted additives adsorbed. The below figure b shows that the Cs⁺ peak in CH₂- and NH-substituted additives are located at the initial position of Cs⁺ at 23.93 Å, which suggests the additives have negligible interactions with the Cs⁺ ion, leaving the Cs⁺ stays at its balance place. However, when the additive has O atoms, the peak representing Cs⁺ ions displaces its original position as compared with the blue dash line. Therefore, we confirm the Tween with O atom plays a key role on changing the optoelectronic properties of Cs-Cu-I system, which is also experimentally proved by introducing a polyether compound of polyethylene oxide (PEO) (Supplementary Fig. 9).

a. The side view in *ab* plane of the charge density difference ($0.0003 e bohr^{-3}$) for CC, CN and CO systems, corresponding to organic molecules using CH₂, NH groups to substitute the O atom in Tween and Tween molecule. The cyan and yellow areas represent electron depletion and accumulation, respectively. The binding energies are 0.023, 0.054 and 0.078 eV, respectively. **b.** The charge density difference along one line passing through the labeled surface Cs atoms for CC, CN and CO systems. The above Cs and line are labeled in the geometric structure scheme.

Comment #4: As authors know, there have been already many reports on the Tween additive for perovskite LEDs. Authors should clarify more the originality of this work.

Response: We thank the reviewer for this suggestion. The Tween additive has been normally used to tune the crystallization of lead-halide perovskites (*Adv. Opt. Mater.* 2018, 6, 1801245). This work found that it can work as a surface modifier to improve the optoelectronic properties of the cesium copper iodides. We have clarified it in the revised manuscript (Line 91 to 92, Page 4, highlighted).

Comment #5: In figure S5b, authors compared the energy levels between w/o Tween and w/ Tween. Ionization potential (I.P.) is intrinsic value of materials. As authors said, if there is difference in I.P., there should be some dipole moment. On the other hand, owing to the dipole moment the conduction band minimum (electron affinity) with Tween become shallow by 0.5 eV, implying that electron injection barrier height from ETL to Cs-Cu-I becomes larger. But, the data for electron only device indicates that electron injection barrier height is similar between w/o Tween and w/ Tween. This point is clearly scientific contradiction.

Response: We thank the reviewer for raising this point. The energy level alignment diagrams of TmPyPB on Cs-Cu-I shows that although the conduction band of Cs-Cu-I with Tween becomes shallow by 0.39 eV (see the below figure a), the electron injection barrier height from ETL (TmPyPB) to Cs-Cu-I is similar (slightly reduced from 1.74 eV to 1.68 eV, see the below figure b), as the Fermi level of the Cs-Cu-I also becomes shallow by 0.45 eV. This is consistent with the data for electron only devices (Supplementary Fig. 6d), which exhibit similar electron injection barrier height.

a, Schematic flat-band energy level diagram. The Fermi level of TmPyPB is from literature (*J. Mater. Chem. C* 2017, 5, 9911-9919). **b**, Energy level alignment diagrams of TmPyPB on Cs-Cu-I without or with Tween.

Reviewer #2:

Comment #1: This submission reports efficient electroluminescence from lead-free copper halide materials, which can be employed to fabricate LED devices. Following the emergence of lead halide perovskite solar cells, metal halides are currently attracting global interest for a variety of optical and electronic applications. In particular, a number of metal halides (both perovskite type and non-perovskite) have recently been shown to exhibit very high efficiency light emission properties. Given the strong interest of a broad community of researchers in luminescent metal halides, the topic of this submission is suitable Nature Communications.

This is an interesting work reporting a substantial improvement of LEDs based on copper iodides. The primary value of this work is the device improvement (i.e., engineering of better devices). I enjoyed reading the paper and I recommend its publication after a revision. In particular, the specific role of additive needs to be clarified better.

Response: We appreciate the reviewer for recognizing the importance of our work and for the constructive comments.

Comment #2: “Among them, solution-processable cesium copper iodides seem promising for white LED applications, because they have high PLQEs with broad visible emission from self-trapped excitons or reorganized excited states, and good air stability.” A few comments here: (i) The authors do not cite a few related papers on cesium copper halides, e.g., [ACS Appl. Electron. Mater. 2019, 1, 3, 269–274] and [ACS Materials Lett. 2019, 1, 4, 459–465]. Air-stability of Cs₃Cu₂I₅ does not seem so good based on a recent publication [ACS Appl. Electron. Mater. 2019, 1, 3, 269–274], any comments? (ii) Do all copper halides demonstrate broad PL emission? Some do not seem so broad, FWHM < 100 nm for powder samples is not so broad? (iii) Some preliminary device work on CsCu₂I₃ has already been reported including in ref. [ACS Materials Lett. 2019, 1, 4, 459–465], in which the device architecture seems to imply the use of organic additives, too. Can the authors compare their results with the literature results and include in Table S1?

Response: We thank the reviewer for this comment. (i) We have cited the related papers on cesium copper halides in the revised manuscript (*ACS Appl. Electron. Mater.* 2019, 1, 269–274; *ACS Materials Lett.* 2019, 1, 459–465). Although the Cs₃Cu₂I₅ exhibits poorer air-stability than

$\text{Cs}_3\text{Cu}_2\text{Br}_5$, it is still much better than lead halide perovskite. We have clarified this in the revised manuscript (Line 57, Page 3, highlighted). (ii) We agree that there are variations in the FWHM of cesium copper iodides, e.g. $\text{Cs}_3\text{Cu}_2\text{I}_5$ (99 nm, *ACS Appl. Electron. Mater.* 2019, 1, 269–274) and CsCu_2I_3 (~110 nm, *Angew. Chem.* 2019, 131, 16233–16237). We have revised the description in the revised manuscript (Line 56, Page 3, highlighted). (iii) We have included the work in Table S1.

Comment #3: In Fig. 1b (also Fig. S2b), why the XRD pattern range is only between 20-30 deg? Also, what are the film thicknesses here, the diffraction peak intensities seem quite low?

Response: We thank the reviewer for this comment. We have updated the XRD figure in the revised manuscript. The thickness of metal halide film is only ~70 nm and it is discontinuous, which can be the reason for the relatively low intensities of diffraction peaks.

a. Adopted from Figure 1. *b.* XRD pattern of film with Tween compared to the calculated XRD patterns of $\text{Cs}_3\text{Cu}_2\text{I}_5$ and CsCu_2I_3 . * and # denote the diffraction peaks corresponding to $\text{Cs}_3\text{Cu}_2\text{I}_5$ and CsCu_2I_3 . *b.* Adopted from Supplementary Figure 2. *b.* XRD pattern of film without Tween.

Comment #4: In Fig. 1e, the film coverage seems poor. What are the darker spots between the grains? If these are Tween formations, is the primary role of Tween then prevention of formation of pinholes? The current explanation of the role of Tween seems to be focused on improved transport properties, which can be a side effect, too (i.e., if film coverage is poor, transport will also be poor).

Response: We thank the reviewer for this comment. As shown in the below figure a, the darker spot between the grains is a ~9 nm organic insulating layer (Tween formations). We agree that Tween plays a role of preventing the formation of pinholes, which can reduce the leakage current. Actually,

the device based on the film without Tween has a higher film coverage (see the below figure b,c), so we believe that the leakage current may not be an main issue for the without Tween devices. We note that the Tween additive significantly increases the injection current, which has been a major issue of cesium-copper-halide LEDs (see the below figure c). Therefore, we believe that Tween plays the role of both prevention of pinholes and enhancement of charge transport properties, and we believe the later is more important. We have clarified the description in the revised manuscript (Line 100 to 101, Page 5, highlighted).

a, Adopted from Supplementary Figure 3. Cross-sectional STEM image of the metal halide film (Scale bar, 100 nm). The STEM measurement was performed on a sample with structure of ITO/PEDOT:PSS/metal halides/Au. The layers in the right panel were tinted. Tween lies between the discrete cesium-copper-iodide crystallites, with a thickness of ~9 nm. **b**, SEM image. Scale bar, 1 μm . **c**, Dependence of current density on the driving voltage.

Comment #5: The exact role of Tween remains confusing in this work. The authors seem to argue about a strong interaction between Tween and cesium copper iodides. What are the HOMO and

LUMO energies of Tween and how do they compare to HOMO/LUMO (VB/CB) of $\text{Cs}_3\text{Cu}_2\text{I}_5$ and CsCu_2I_3 ? If the transport properties are improved, are the authors arguing that Tween helps with charge transport? If yes, please, clarify. Or is it simply a passivation of surface defects with Tween that helps the device performance?

Response: We thank the reviewer for this comment. The Tween additive is a large band-gap insulator, which should have much lower HOMO and higher LUMO than those of $\text{Cs}_3\text{Cu}_2\text{I}_5$ and CsCu_2I_3 . Thus it can not be regarded as charge-transport layer but rather as surface modifier. The scanning Kelvin probe microscope (SKPM) and ultraviolet photoelectron spectroscopy (UPS) measurements indicate that Tween can increase the surface potential of cesium-copper-halide film and reduce the energetic barrier for hole injection (Fig. 3 and Supplementary Fig. 6a). The density functional theory calculation confirms that Tween can modify the crystallographic facets of CsCu_2I_3 and $\text{Cs}_3\text{Cu}_2\text{I}_5$ through the interaction between oxygen and Cs ions (Supplementary Fig. 7a,b), which further reduces the work function and enhances the charge transport of cesium-copper-iodide layer. The calculation result is consistent with the measurement result of the single-carrier devices, which shows that the inclusion of Tween enhances the current density in electron- and hole-only devices (Supplementary Fig. 6c,d). We have clarified the description in the manuscript (Line 198 to 202, Page 8).

Comment #6: The current best device seems to be mostly yellow emitting CsCu_2I_3 and the reported CIE coordinates seem further away from the white point. Can the authors optimize $\text{Cs}_3\text{Cu}_2\text{I}_5/\text{CsCu}_2\text{I}_3$ ratio to get closer to the white point? And more generally, how does changing the $\text{Cs}_3\text{Cu}_2\text{I}_5/\text{CsCu}_2\text{I}_3$ ratio impact the EQE of the resultant LEDs?

Response: We thank the reviewer for this comment. The below figure a shows that changing the $\text{Cs}_3\text{Cu}_2\text{I}_5/\text{CsCu}_2\text{I}_3$ ratio can tune the PL spectra of resultant films. The CsI:CuI (2.5:2) film reaches a CIE color coordinate of (0.32, 0.32), exhibiting white emission (see the below figure b). However, we note that changing the ratio of $\text{Cs}_3\text{Cu}_2\text{I}_5$ and CsCu_2I_3 will significantly reduce the EQE of device (see the below figure c,d,e). This low device efficiency could be mainly due to the poor film morphology (see the below figure g). Further device engineering work is ongoing. We have added the data in the revised manuscript (Supplementary Figure 5, highlighted).

Reviewer #3:

Comment #1: The authors report the use of the Tween additive in Cs-Cu-I and successfully fabricate white LEDs with record performance in the lead-free halide family. Based on XRD, SEM, FTIR and GIWAXS, detailed analysis and calculations were carried out, which help explain the unique optoelectronic properties. This is an interesting and important work on the lead-free halides research.

Response: We thank the reviewer for recognizing the importance of our work and for the constructive comments.

Comment #2: The XRD was only measured in a very short range between 20 and 30°. The authors need to measure from 5 to 40° and also include the simulated patterns in Figure 1b for comparison.

Response: We thank the reviewer for this suggestion. We have updated the XRD figure in the revised manuscript. It confirms that the prepared film is comprised of zero-dimensional $\text{Cs}_3\text{Cu}_2\text{I}_5$ and one-dimensional CsCu_2I_3 .

Figure 1. b, XRD pattern compared to the calculated XRD patterns of $\text{Cs}_3\text{Cu}_2\text{I}_5$ and CsCu_2I_3 . * and # denote the diffraction peaks corresponding to $\text{Cs}_3\text{Cu}_2\text{I}_5$ and CsCu_2I_3 .

Comment #3: From the SEM image, the thin film looks like isolated $\text{Cs}_3\text{Cu}_2\text{I}_5$ and CsCu_2I_3 were formed. In this case, do the LED devices have current leakage in the longitudinal direction? This requires careful consideration and discussions in the manuscript.

Response: We thank the reviewer for this comment. As shown in the below figure a, there is a ~9 nm organic insulating layer between the discrete cesium-copper-halide crystallites of film with Tween

additive, which can suppress leakage current. In addition, the J - V curves show that the addition of Tween reduces the leakage current of LED device (below figure b). We have clarified the description in the revised manuscript (Line 100 to 101, Page 5, highlighted).

a, Adopted from Supplementary Fig. 3. Cross-sectional STEM image of the metal halide film (Scale bar, 100 nm). The STEM measurement was performed on a sample with structure of ITO/PEDOT:PSS/metal halides/Au. The layers in the right panel were tinted. Tween lies between the discrete cesium-copper-iodide crystallites, with a thickness of ~ 9 nm. **b**, Dependence of current density on the driving voltage of devices with or without Tween.

Comment #4: The authors claim that the film is very stable in air (over 1500h). How stable are the LED devices?

Response: The LED device reaches a half-lifetime of 230 s at an initial luminance of ~ 100 cd m^{-2} (see the below figure). The modest stability of LED devices is probably due to the deterioration of interfacial properties of the devices.

Stability of the device measured at a constant current density of 3 mA cm⁻².

Comment #5: The white emission should be caused by the different photoluminescence behaviors of Cs₃Cu₂I₅ and CsCu₂I₃. Have the authors tried films with different ratios of CsI and CuI? If the PL changes with different starting materials, it will confirm the origin of the white light.

Response: We thank the reviewer for raising this point. We confirm the white emission stems from the different PL behaviors of Cs₃Cu₂I₅ and CsCu₂I₃. The below figure shows that the CsI:CuI (1:2) film has a main emission peak at 570 nm and a shoulder at 408 nm, corresponding to the excited-state structural reorganization emission of CsCu₂I₃ and CuI, respectively (*Angew. Chem.* 2019, 131, 16233–16237; *Chem. Lett.* 2005, 34, 1158–1159). The CsI:CuI (3:2) film has a main emission peak at 437 nm from the self-trapped exciton emission of Cs₃Cu₂I₅. Consequently, the CsI:CuI (1:1) film exhibits a dominant PL peak from CsCu₂I₃ at 570 nm with an emission shoulder from Cs₃Cu₂I₅ at 437 nm, which leads to the warm-white emission. We have added the data in the revised manuscript (Supplementary Fig. 5a).

PL spectra of metal halide films with different ratios of CsI:CuI.

Comment #6: I'm concerned about the statement of the role of Cs⁺ and Tween. Compared with Cs, Cu element has greater electronegativity. Tween should also play a role on Cu site.

Response: We thank the reviewer for this comment. We agree that Tween also can interact with Cu site in principle. However, the exposure of (010) surface terminated Cu atoms is more unstable than Cs atoms. The calculations show that the surface energies for Cs-terminated (010) surface and

Cu-terminated (010) surface are $0.453 \text{ eV \AA}^{-2}$ and $5.564 \text{ eV \AA}^{-2}$ respectively, suggesting the preferred interaction between Tween and Cs site. We have clarified the description in the revised manuscript (Line 430 to 434, Page 21, highlighted).

REVIEWER COMMENTS

Reviewer #1 (Remarks to the Author):

I carefully checked the authors' answers. I think there remain several issues. Please see below.

1. Authors said that "The energy level alignment diagrams of TmPyPB on Cs-Cu-I shows that although the conduction band of Cs-Cu-I with Tween becomes shallow by 0.39 eV (see the below figure a), the electron injection barrier height from ETL (TmPyPB) to Cs-Cu-I is similar (slightly reduced from 1.74 eV to 1.68 eV, see the below figure b), as the Fermi level of the Cs-Cu-I also becomes shallow by 0.45 eV." However, this explanation based on Fermi level shift is wrong. For the interface between organic and inorganic materials, thermal equilibrium does not occur since there is no free carrier in organic layer. This explains why only electron affinity level or ionization potential is considered for charge injection barrier height in organic electronics such as OLEDs; only the highly-doped organic semiconductors such as PEDOT:PSS can possess band alignment based on thermal equilibrium.

2. Authors said that "The Tween additive has been normally used to tune the crystallization of lead-halide perovskites (Adv. Opt. Mater. 2018, 6, 1801245). This work found that it can work as a surface modifier to improve the optoelectronic properties". This answer is very disappointing. Authors definitely read the paper, Adv. Opt. Mater. 2018, 6, 1801245. In the conclusion of the paper, Liu et al. said "A nonionic surfactant Tween 20 was introduced into the all-inorganic CsPbBr₃ emission layer, significantly enhancing the PLQY by passivating the defects and traps at the grain boundaries." Moreover, they well showed how the PL lifetime become longer with Tween additive by decreasing trap or defect states.

3. For XPS data in Fig. 4b, peak shift of Cs 3d cannot be the evidence of Cs-Tween interaction. According to the Fermi energy for w/Tween, workfunction is ~0.5 eV smaller than w/ Tween. In this case, B.E. should be larger (see the attached figure). On the contrary, authors said that Cs 3d binding energy for w/ Tween became smaller by ~1.6 eV, implying that real chemical state difference between w/ and w/o Tween is ~2.1 eV. The difference of ~2.1 eV is the same value between Cs⁺ and Cs⁰ states. Is that scientifically possible? Authors should show whether there is also change in Cu and I peaks.

In conclusion, I feel that authors discuss only what they want. There are still many unclear issues. Of course, I agree that authors well studied the effect of Tween in Cs-Cu-I system. However, similar phenomenon was already reported in Adv. Opt. Mater. 2018, 6, 1801245. In this case, authors had to find more new things than the preceding study. From this view point, it is rather important work to clarify how the Tween works well in halides, but, I do not think authors accomplished the task. I feel there is no new discovery in this paper. I agree that precise analyses is quite difficult in multilayered devices such as ELs. Sometimes, we experience many unclear phenomena. In this case, overclaiming is not necessary. Of course, I agree that authors well applied the Tween to Cs-Cu-I ELs and authors also tried to study the role of Tween. But, this paper should be improved a lot to satisfy the standard of Nature communication. Please carefully check my comments again and revise the manuscript.

Reviewer #2 (Remarks to the Author):

In their revised submission, the authors have satisfactorily addressed all of my critical concerns. Upon a review of other referee questions and concerns, I find their responses to other question to be satisfactory as well. Therefore, I am now happy to recommend this work for publication

Reviewer #3 (Remarks to the Author):

Generally, the authors well addressed my comments. I have two more suggestions which the authors should answer:

(1) Now the XRD pattern has the simulated patterns for comparison. However, the measured result is too rough to be confirmed. The authors need to carry out a slow scan or a one with stronger X-ray source for measurement. Right now, the statement about the two phases based on the XRD result is not convincing.

(2) For the interaction between Cu and Tween, the authors replied with a result based on calculation, which may be acceptable. If an experiment data could be provided, it would be great. If not available, a statement about the potential strong interaction between Cs and Tween could be added, which could help give promising directions towards the future deep analysis.

Point-by-Point Response to Referees

Reviewer #1:

Comment #1: I carefully checked the authors' answers. I think there remain several issues. Please see below.

In conclusion, I feel that authors discuss only what they want. There are still many unclear issues. Of course, I agree that authors well studied the effect of Tween in Cs-Cu-I system. However, similar phenomenon was already reported in *Adv. Opt. Mater.* 2018, 6, 1801245. In this case, authors had to find more new things than the preceding study. From this view point, it is rather important work to clarify how the Tween works well in halides, but, I do not think authors accomplished the task. I feel there is no new discovery in this paper. I agree that precise analyses is quite difficult in multilayered devices such as ELs. Sometimes, we experience many unclear phenomena. In this case, overclaiming is not necessary. Of course, I agree that authors well applied the Tween to Cs-Cu-I ELs and authors also tried to study the role of Tween. But, this paper should be improved a lot to satisfy the standard of Nature communication. Please carefully check my comments again and revise the manuscript.

Response: We thank the reviewer for the helpful and constructive comments. We have revised the manuscript accordingly. We would like to point out that in this work, beside reducing defect states, the Tween additive play roles of enhancing crystallinity, charge transfer and increasing surface potential through the interaction between ether groups and Cs ions, which are fundamentally different from the lead system in the literature (*Adv. Opt. Mater.* 2018, 6, 1801245) and are completely new in the lead-free halide field. Nevertheless, the most important thing with the work is the demonstration of good performance lead-free halide LEDs.

Comment #2: Authors said that "The energy level alignment diagrams of TmPyPB on Cs-Cu-I shows that although the conduction band of Cs-Cu-I with Tween becomes shallow by 0.39 eV (see the below figure a), the electron injection barrier height from ETL (TmPyPB) to Cs-Cu-I is similar (slightly reduced from 1.74 eV to 1.68 eV, see the below figure b), as the Fermi level of the Cs-Cu-I also becomes shallow by 0.45 eV." However, this explanation based on fermi level shift is wrong. For the interface between organic and inorganic materials, thermal equilibrium does not occur since

there is no free carrier in organic layer. This explains why only electron affinity level or ionization potential is considered for charge injection barrier height in organic electronics such as OLEDs; only the highly-doped organic semiconductors such as PEDOT:PSS can possess band alignment based on thermal equilibrium.

Response: We thank the reviewer for this comment. First of all, we agree that the contacts between organic and inorganic materials are complex, usually involving interface dipole and band bending. However, there are many evidences showing that the Fermi level alignment does occur in various organic semiconductors (for example: *Phys. Stat. Sol. (a)* 201, 1075 (2004); *Phys. Rev. Lett.* 106, 216402 (2011); *Org. Electron.* 13, 1680 (2012); *Phys. Rev. B* 90, 045302 (2014); *Phys. Rev. B* 90, 045303 (2014)). In our case, the experimental result (Supplementary Fig. 6) shows that both the hole current and electron current are enhanced with the Tween additive, and it seems that the Fermi level alignment can explain the result well. We have added this in the revised manuscript “*A schematic flat-band energy level diagram for our devices is shown in Supplementary Fig. 6b, although there are likely complex band-bending and interactions that will occur close the interfaces. However, the diagram is able to show that Tween can reduce the energetic barrier for hole injection owing to the shallow valence-band energy level of emitting layer.*” (Line 142 to 146, Page 6, highlighted).

Comment #3: Authors said that “The Tween additive has been normally used to tune the crystallization of lead-halide perovskites (*Adv. Opt. Mater.* 2018, 6, 1801245). This work found that it can work as a surface modifier to improve the optoelectronic properties”. This answer is very disappointing. Authors definitely read the paper, *Adv. Opt. Mater.* 2018, 6, 1801245. In the conclusion of the paper, Liu et al. said “A nonionic surfactant Tween 20 was introduced into the all-inorganic CsPbBr₃ emission layer, significantly enhancing the PLQY by passivating the defects and traps at the grain boundaries.” Moreover, they well showed how the PL lifetime become longer with tween additive by decreasing trap or defect states.

Response: We thank the reviewer for this comment. We have added the description of passivation effect of Tween from the literature in the revised manuscript (Line 95, Page 4, highlighted). We would like to state that our work found some fundamentally different effects of Tween in Cs-Cu-I through the interaction between ether groups and Cs ions: 1) The synchrotron-based *in-situ* GIWAXS

measurements show that Tween can retard the nucleation of cesium copper iodides, leading to film with enhanced crystallinity. 2) The SKPM and UPS measurements show that Tween can increase the surface potential of cesium copper iodides, which is beneficial for hole injection. 3) The characterization of single carrier devices demonstrates that Tween can enhance the charge transport in device. 4) The DFT calculation result is consistent with the experimental observations.

Comment #4: For XPS data in Fig. 4b, peak shift of Cs 3d cannot be the evidence of Cs-Tween interaction. According to the fermi energy for w/Tween, work function is ~0.5 eV smaller than w/o Tween. In this case, B.E. should be larger (see the attached figure). On the contrary, authors said that Cs 3d binding energy for w/ Tween became smaller by ~1.6 eV, implying that real chemical state difference between w/ and w/o Tween is ~2.1 eV. The difference of ~2.1eV is the same value between Cs⁺ and Cs⁰ states. Is that scientifically possible? Authors should show whether there is also change in Cu and I peaks.

Response: We thank the reviewer for this very helpful comment. Indeed, the XPS result was not accurate. We have remeasured XPS spectra of samples on PEDOT:PSS substrates and carried out the sample loading in inert atmosphere, which can keep the same structure as device and reduce the influence of adsorbed oxygen or other contaminations. The new result shows that the Cs 3d and I 3d binding energy for Tween:CsI sample became smaller by ~0.3 eV compared with CsI, while the peak associated with O 1s in Tween moves to higher binding energy upon the addition of CsI. Similar results are also observed in CsI:CuI sample with Tween. On the contrary, the inclusion of Tween causes no chemical shift in the CuI sample. This result is consistent with our calculation data, which show that the electron transfer occurs heavily at the interface between Cs and I atoms, and it transfers from the organic to inorganic part through O atoms (Line 177 to 194, Page 7 and Line 446 to 447, Page 21). We have updated the results in the revised manuscript (Figure 4b, Supplementary Figure 7, Line 169 to 176, Page 7, highlighted). In addition, our FTIR measurement also can confirm the chemical interaction between Cs and the C-O-C bond of Tween (Fig. 4a, Line 162 to 167, Page 7).

XPS spectra. *a*, Cs 3d and I 3d spectra of CsI, Tween:Csl, CsI:Cul and Tween:Csl:Cul films. *b*, Cu 3d and I 3d spectra of CuI, Tween:Cul, CsI:Cul and Tween:Csl:Cul films. *c*, O 1s spectra. The O 1s lines were decomposed into two or three peaks. The red lines represent the raw data, while the blue, green lines represent the PEDOT:PSS and the purple lines represent the Tween. It shows that the peak of C=C-O group in PEDOT:PSS located at ~531.5 eV (*ACS Appl. Mater. Interfaces* 2014, 6, 2292–2299; *Adv. Energy Mater.* 2017, 7, 1602116) shifts to 531.9 eV in the samples with CsI or CuI, which indicates that the PEDOT:PSS substrate has a strong coupling with CsI, CuI, and CsI:Cul films. But the Tween additive can reduce this interaction, which shows the corresponding C=C-O peak locating at ~531.5 eV. The peak at ~532.8 eV assigned to the oxygen of Tween has no shift after the inclusion of CuI, but shifts to higher binding energy in Tween:Csl sample.

Reviewer #2:

Comment #1: In their revised submission, the authors have satisfactorily addressed all of my critical concerns. Upon a review of other referee questions and concerns, I find their responses to other question to be satisfactory as well. Therefore, I am now happy to recommend this work for publication.

Response: We thank the reviewer for the positive comment.

Reviewer #3:

Comment #1: Generally, the authors well addressed my comments. I have two more suggestions which the authors should answer.

Response: We thank the reviewer for the positive comment.

Comment #2: Now the XRD pattern has the simulated patterns for comparison. However, the measured result is too rough to be confirmed. The authors need to carry out a slow scan or a one with stronger X-ray source for measurement. Right now, the statement about the two phases based on the XRD result is not convincing.

Response: We thank the reviewer for this suggestion. We have carried out slow XRD measurements and updated the XRD figures (Fig. 1b and Supplementary Fig. 2b) in the revised manuscript. It clearly shows that the prepared film is comprised of zero-dimensional $\text{Cs}_3\text{Cu}_2\text{I}_5$ and one-dimensional CsCu_2I_3 .

Comment #3: For the interaction between Cu and Tween, the authors replied with a result based on calculation, which may be acceptable. If an experiment data could be provided, it would be great. If not available, a statement about the potential strong interaction between Cs and Tween could be added, which could help give promising directions towards the future deep analysis.

Response: We thank the reviewer for this comment. We have added the statement about the potential strong interaction between Cs and Tween in the revised manuscript (Line 446 to 447, Page 21, highlighted).

REVIEWERS' COMMENTS

Reviewer #1 (Remarks to the Author):

Authors responded to all my concerns and well revised the manuscript. I suggest the publication of this paper in Nature communications.

Reviewer #3 (Remarks to the Author):

The authors have addressed my comments about XRD and the interaction between elements. Now I think it's available for publication.

Point-by-Point Response to Referees

Reviewer #1:

Comment #1: Authors responded to all my concerns and well revised the manuscript.

I suggest the publication of this paper in Nature communications.

Response: We thank the reviewer for the positive comment.

Reviewer #3:

Comment #1: The authors have addressed my comments about XRD and the interaction between elements. Now I think it's available for publication.

Response: We thank the reviewer for the positive comment.